# A Mixed-Methods Systematic Review of Group Reflective Practice in Medical Students

**DOI:** 10.3390/healthcare11121798

**Published:** 2023-06-19

**Authors:** Kelvin C. Y. Leung, Carmelle Peisah

**Affiliations:** 1Research and Education Network, Westmead Hospital, Sydney, NSW 2145, Australia; 2Discipline of Psychiatry and Mental Health, Faculty of Medicine, University of New South Wales, Sydney, NSW 2052, Australia; 3Specialty of Psychiatry, Faculty of Medicine and Health, University of Sydney, Sydney, NSW 2006, Australia

**Keywords:** reflective practice, continuing medical education, empathy, professionalism, wellbeing

## Abstract

Background: Used primarily as a pedagogical evaluation tool for didactic teaching and skill development, reflective practice (RP) for its own merits is poorly understood. This study aimed to systematically review the literature regarding the role of group RP in fostering empathy, wellbeing, and professionalism in medical students. Methods: Electronic searches of empirical studies published between 1 January 2010 and 22 March 2022 from Medline, Embase, and PsychINFO databases were conducted. Empirical studies of any design (qualitative or quantitative) which included RP (1) involving medical students; (2) with a focus on fostering empathy, or professionalism, or personal wellbeing; and (3) provided in a group setting were included. Duplicates, non-English articles, grey literature and articles using RP to examine pedagogy and specific technical skills were excluded. Both authors screened articles independently to derive a final list of included studies, with any discrepancies resolved by discussion, until consensus reached. Articles were rated for methodological quality using the Attree and Milton checklist for qualitative studies; the Oxford Centre for Evidence-Based Medicine criteria, and the Alberta Heritage Foundation for Medical Research Standard Quality Assessment Criteria for quantitative studies. Results: Of 314 articles identified, 18 were included: 9 qualitative; 4 quantitative and 5 mixed methodology. Settings included United States (6), United Kingdom (3), Australia (3), France (2), Taiwan (2), Germany (1), and Ireland (1). Themes were (i) professionalism: bridging theoretical paradigms and practice; (ii) halting empathy decline; (iii) wellbeing: shared experience. Additional themes regarding the “successful“ delivery of RP groups in facilitating these outcomes also emerged. Conclusions: This first systematic review of group RP in medical students shows that RP may bring theory to life in clinical dilemmas, while fostering collegiality and mitigating against isolation amongst students, despite the absence of studies directly examining wellbeing. These findings support the value of RP integration focusing on emotive and humanitarian processes into contemporary medical education for medical students. Systematic review registration: PROSPERO CRD42022322496.

## 1. Introduction

Reflective practice (RP) has long-been a tool in medical education [1]. With broad origins across sociology, management and organisational learning dating back to the 1930s, the term RP lacks clarity in definition [2]. This lack of clarity is the culmination of almost a century of an evolving RP paradigm based on a range of learning experiences involving both cognitive processes (i.e., thinking about the experience) and emotive processes (i.e., feeling about the experience). This fittingly explains its inherent and diverse use in medicine. For example, as applied to medicine, RP may be defined as a process of self-questioning and experiential learning involving recapturing and evaluating clinical experiences to promote self-understanding, professionalism, “practical wisdom” and ideally, patient-care [3]. At its essence, the use of RP is the application of experiential learning to inform and influence future outcomes and practices [2]. As such, in the context of medical practices, these outcomes relate to, and ideally benefit, both patients and doctors alike.

Best undertaken as guided or facilitated [1], group RP is an efficient mode for facilitated delivery. Notably, key to many of the original conceptualisations of RP was that it was based on individual learning through a collective means, namely by observation of, and learning from, others [2]. The use of group facilitation of RP is exemplified by the Balint group, a particular type of reflective practice where clinicians meet regularly to discuss cases from their practices, with a focus on emotional content and clinical interactions [4].

Intuitively, therefore, practising reflection early and nurturing it longitudinally throughout one’s medical career would appear to be particularly useful in regards to fostering professionalism and the humanitarian aspects of medicine (e.g., respect for others, integrity, duty, honour, altruism, accountability and excellence) [5]. The importance of bringing these “soft skills” out of the hidden curriculum of medical training and into the open curriculum has not been lost on medical educators [5]. Notwithstanding these instinctive benefits of RP, actual empirical investigation of the benefits of RP for fostering humanitarianism is extremely limited. Notably, a redesigned integrative curriculum incorporating group RP at Harvard Medical School has shown a long-lasting improvement in psychosocial, relational and humanistic attitudes in medical graduates, even when confidence in managing patients with psychosocial problems and practising humanistic medicine was evaluated 10 years later [6].

Beyond this curriculum examination, RP has been evaluated only as a pedagogical evaluation tool, for example, to ask learners the question: how did you learn communication skills [7], first-year pathology [8], case-solving skills [9] or procedures [10]? While the use of RP for such technical skill acquisition is an important learning process for medical students, we know little about its use in fostering more emotive, abstract processes intrinsic to medical professionalism. This unexamined use is also more aligned with its original intended purpose [2]. In particular, the role of RP in promoting professionalism, empathy, as well as emotional self-benefit, have not been reviewed. The latter benefit is of particular relevance in the wake of the COVID-19 pandemic and its effects on wellbeing of medical students [11,12], who are often neglected in the wellbeing space. Further to our earlier comments about lifelong learning starting in medical school, and as deliverers ourselves of an RP programme for medical students, we sought to examine the evidence in this population.

This highlights an untapped evidence-base regarding the value of RP beyond its use to augment technical skill-based learning. This systematic review aims to explore the literature regarding the use of group-based RP to enhance empathy, professionalism and personal wellbeing amongst medical students. The modified research PICo (Population, Interest, Context) question [13] is: *What is the role of group reflective practice in promoting empathy, professionalism and wellbeing in medical students?*

## 2. Materials and Methods

### 2.1. Design, Protocol, and Registration

Preferred Reporting Items for Systematic Reviews and Meta-Analysis (PRISMA) recommendations were used as a framework for this review [14]. The protocol for this systematic review was registered on PROSPERO (ID: CRD42022322496).

### 2.2. Search Strategy

Electronic searches of databases, including Medline, Embase, and PsychINFO, between 1 January 2010 and 22 March 2022 were conducted using search terms designed to identify studies reporting the use of any forms of group RP in medical students. The restriction to 2010 was chosen because we aimed to capture approaches which reflect contemporary educational practice and context.

Search terms were determined using an iterative process by identifying common terminology used in the literature to cover the three key domains (empathy, professionalism and wellbeing) in our research question. For professionalism, despite its lack of clarity in definition, we used concepts of ‘clinical competence’, ‘patient-centred care’ and ‘doctor-patient relationship’ in our search terms as they were the most recursive in our preliminary searches. In terms of operationalising wellbeing, we utilised the most studied construct: ‘burnout’, as a search proxy for wellbeing.

The following combination of terms were used:‘reflection’, or ‘reflective practice’, or ‘reflective thinking’, or ‘reflective learning’, or reflective group’, or ‘balint group’, AND‘group’, AND‘medical students’, AND‘empathy’, or ‘clinical competence’, or ‘patient centred care’, or ‘communication’, or ‘doctor patient relationship’, or ‘burnout’.

### 2.3. Study Selection

#### 2.3.1. Inclusion Criteria

The published peer-reviewed literature was reviewed. Empirical (original) studies, of any design (qualitative or quantitative) which included RP (1) involving medical students; (2) with a focus on fostering empathy, or professionalism, or personal wellbeing; and (3) provided in a group setting, were identified.

#### 2.3.2. Exclusion Criteria

Excluded articles included commentaries, literature reviews, meta-analyses, editorials, letters or grey literatures. Articles were also excluded if (1) the content did not have a component of reflection; (2) RP was used as a pedagogical evaluation tool for medical education, curriculum or didactic programme (e.g., improving delivery and/or design of these programmes); focused on specific technical skill acquisition (e.g., procedural skills, communication skills, clinical reasoning, and diagnostic competence); (3) RP was undertaken as a solo exercise; or (4) they were non-English articles.

### 2.4. Review Team

The review team comprised experienced systematic reviewers, with qualitative and quantitative research experience, as well as being content experts responsible for delivering RP groups. The lead author, an advanced psychiatry trainee with expertise in delivering RP groups and undertaking systematic reviews, undertook the database searches, screening and integration. The second author, who is a senior psychiatrist with extensive experience undertaking qualitative research and systematic reviews as well as delivering RP groups, assisted with screening and the thematic analysis.

### 2.5. Screening and Data Extraction

Database searches were performed, validated and short-listed by the first author. The short-listed abstracts were screened by both authors to determine their eligibility against the inclusion and exclusion criteria. Any disagreements were discussed to reach consensus. For abstracts meeting inclusion criteria, full-text articles were then obtained for further screening performed by both authors working independently to derive a final list of included studies, with any discrepancies resolved by discussion until consensus reached.

### 2.6. Quality Assessment

Qualitative studies were rated using Attree and Milton’s checklist (2006) [15]. This checklist included criteria for rating methodological quality such as research context and background, aims and objectives, study design, sampling, data collection, results analysis, reflexivity, study value and ethical considerations. Each checklist domain was rated from A (no or few flaws) to D (significant flaws threatening the study validity), with the final quality score (A–D) determined by majority grade across domains.

Quantitative studies were appraised using Kmet et al.’s Alberta Heritage Foundation for Medical Research Standard Quality Assessment Criteria (2004) [16]. The checklist provided operationalised criteria including objectives, appropriateness of design, selection of subjects, random allocation and blinding, exposure and outcome measures, sample size, analytic methods, estimates of variance, confounding, results reporting and conclusions, with a final rating score expressed as percentage of the maximum total score. While there is no established score-based rating for overall quality, other systematic reviews have defined >80% as high quality [17,18].

Levels of evidence for quantitative studies were rated using the Oxford Centre for Evidence-Based Medicine criteria (2011) [19]. For interventional studies, level 1 includes systematic reviews of randomised or n-of-1 trials; level 2 includes randomised trials and observational studies with dramatic effect; level 3 includes cohort studies; level 4 includes case-controlled studies, case series, or historically controlled studies; and level 5 is mechanistic reasoning. Level may be graded down based on methodological flaw or small effect size. Qualitative studies are not considered for these criteria.

Both authors independently scored all included papers for quality according to the above criteria with scoring differences discussed until consensus reached.

### 2.7. Data Analysis

A table was created to extract relevant data, including author details, year of studies, country of studies, study aims, characteristics of participants and settings, study design, comparison group(s) (if any), outcome measures, limitations, level of evidence and aspects of methodological quality and score (see Appendix A). Both authors reviewed the data synthesis.

The heterogeneity of studies and inconsistent use of measures meant that the data collected in this review were unsuitable for quantitative synthesis or a meta-analysis [20].

We used a convergent integrated approach for this mixed-methods systematic review, combining both forms of data (i.e., qualitative and quantitative) into a single mixed methods synthesis, codifying both forms of data using thematic analysis. As such, data was synthesised qualitatively using inductive thematic analysis to identify salient themes. Differences and similarities across the data set were revealed using an iterative, constant comparison method [21]. First, the data was coded separately by both authors, looking for emerging themes from the included papers. Second, both authors discussed the codes and jointly re-coded potentially unclear ones. Reflexivity was considered at every step from data collection to thematic analysis. It is worthy to note that, as previously stated, both authors were involved in the delivery of medical student RP groups, and, as champions of the technique, were perhaps positively favoured towards its delivery and its justification. Finally, all findings were critically tested and discussed to resolve any discrepancies.

## 3. Results

### 3.1. Search Results

From an initial 314 records identified, 18 studies were included in the review, as per the PRISMA flow diagram (Figure 1).

### 3.2. Study Characteristics

Of 18 studies included, six were from United States [22,23,24,25,26,27], three from United Kingdom [28,29,30], three from Australia [31,32,33], two from France [34,35], two from Taiwan [36,37], one from Germany [38] and one from Ireland [39].

Nine used qualitative methodology [23,27,29,31,32,33,36,37,38], four used quantitative methodology [26,30,34,35] and five used mixed methodology [22,24,25,28,39].

Study settings were all based in universities or medical schools from metropolitan or urban areas in which students were doing their clinical years, except for Gold et al. (2019) [25], which recruited first- and second-year medical students, whose clinical experiences were unknown. 12 studies included RP groups integrated as part of a clerkship, curriculum, or certificate [22,23,26,27,32,33,36,39], while 8 others involved formal delivery of RP groups as a trial either on its own or as part of a program [24,25,28,29,30,31,34,35,38].

The characteristics, summary findings, quality ratings, and level of evidence for each included study are summarised in Table A1 (see Appendix A).

### 3.3. Quality and Bias Analysis

Of 14 studies utilising qualitative methodology, three were rated ‘A’ [27,28,38]; six were rated ‘B’ [24,29,33,36,37,39]; and five were rated ‘C’ [22,23,25,31,32] using Attree and Milton’s ratings (2006).

Of the nine studies using quantitative methodology [22,24,25,26,28,30,34,35,39], quality ratings ranged from 27% to 93% using Kmet et al.’s criteria (2004). Level of evidence based on OCEBM (2011) ratings included two at level 2 [34,35]; five at level 3 [22,24,26,28,30]; and one each at levels 4 [39] and 5 [25].

### 3.4. Synthesis of Results

The thematic analysis generated a number of themes elucidating links between group RP and professionalism, empathy and wellbeing in medical students:

#### 3.4.1. Professionalism: Bridging Clinical and Theoretical Paradigms to Serve the Doctor Patient-Relationship

Six of the studies demonstrated that RP cultivates professionalism in medical students by bridging theory and practice in relation to the doctor-patient relationship [29] and the biopsychosocial context [36]. Often in relation to dilemmas in clinical practice and complex patients [27], reflections can be triggered specifically in relation to older patients [28], patients with borderline personality disorder [34], and clerkship challenges [23]. These studies illustrated the value of RP in facilitating the emotional component of patient interactions and the humanitarian aspects of professionalism. What was notable was the role of formal structured RP, namely Balint groups, in educating students about these emotional aspects of the doctor-patient relationship [29,34].

In a RP group, students can bring in complex cases and clinical dilemmas [27], often not being immediately clear to them the nuance of the patient contact, clinical context, and/or psychosocial factors, which may interplay with their emotional and personal experience in relation to these cases and beyond [23,25,29,36]. Moreover, Bird et al. emphasised the importance of creating a setting that is conducive to “comfortable reflection” [22]. These cumulative elements appear to be key to assist medical students make sense of the theoretical bases behind their clinical encounters and contextualise these humanistic interactions. Five studies specifically addressed that having awareness to the doctor-patient relationship is akin to medical professionalism [28,29,31,32,33].

#### 3.4.2. Empathy: Halting Empathy Decline

Six studies examined the effect of RP on empathy preservation as demonstrated by Jefferson Scale of Empathy [24,26,35,39], modified Emotional Self-Awareness Scale [25] and the empathic approach in response to case reports [34].

Three studies demonstrated a significant improvement of self-administered empathy scores after RP groups [26,35,39]. A notable finding was that empathy or the ability to tolerate diverse perspectives was perceived to decrease over time [34], but RP groups may have a place in preserving students’ empathy [24], or even improving empathic ability throughout the course of patient care [25]. RP groups may also provide opportunities to contextualize empathic responses. Again, Balint groups may be the key to this [34,35]. One study, in particular, utilised 10 two-hour weekly Balint sessions to facilitate enhanced empathic approaches within the context of the actual doctor-patient relationship, rather than promoting more general empathic responses [34].

#### 3.4.3. Wellbeing: The Value of Shared Experience

No study directly measured wellbeing. However, one RP embedded within a curriculum used RP to promote resilience, helping students deal with setbacks and challenges experienced during clinical training, while also finding meaning [22]. Four studies demonstrated that group practice specifically facilitated connectedness and offset isolation in medical school [22,24,25,37], providing a safe environment for mutual support and shared experiences, as well as allowing exposure to and tolerance of diverse perspectives [25,37].

On the other hand, two studies found that some medical students felt unable to share their voice at times, restrained by feared repercussions of opening up in hierarchical environment [27,38]. This was even in the face of potentially inappropriate or harmful practices observed.

These indirect but potentially positive effects on student wellbeing mediated by RP groups appeared to be contingent upon providing a “comfortable” environment for reflection [22].

As such, other important additional themes emerged from the review in relation to the practical and successful delivery of RP groups:

#### 3.4.4. Ingredients for Successful RP Groups

A number of factors were identified that may either enhance or detract from the “success” of RP groups. For example, the various benefits of voluntary [22,29,31] versus compulsory attendance [32,38] have been explored. Voluntary, unforced participation appears to foster a sense of safety and collegiality in the group. These participants may predispose to benefits from the process. However, without the need to attend compulsorily students may rather resort to their familiar learning methods, missing the benefits of group RP. The challenges of making reflection relevant to medical students were reported, with some students failing to see any relevance from such reflection for either their work as students or physicians [22,32,38], suggesting that the utility of such groups may be contingent upon the psychological mindedness of the particular cohort. Brand et al. (2016) explored the value of “pre-reflection” to facilitate delivery and engagement of students [28]. Timing of groups [31], when students have sufficient clinical experience, or specifically, under general practice settings [32,33] to render such groups more meaningful also appears to be essential. Highlighting the importance of safe space [22,24], some students encountered interpersonal problems that impeded openness to engage. Special adaptation of group RP or Balint group to medical students and maintaining a dynamic approach responsive to the needs of a particular cohort are hence key elements. However, a pragmatic component of success is ultimately finding enough enthusiastic and skilled facilitators to run such groups [38].

#### 3.4.5. Innovative Delivery Methods

A range of innovative RP delivery methods have been examined including (i) the Depth of Field reflective learning resources which uses photo-elicitation techniques, older adults’ narratives and collaborative dialogues in the classroom [28]; (ii) resilience skills curriculum which employs core virtues of positive psychology, including intellectual strengths, interpersonal strengths, and temperance strengths [22]; (iii) the VALOR program which involves peer groups of balanced demographics such as gender, preferences for clerkship order, and prior experiences with other students in the cohort [23]; and (iv) RP embedded within the Student Psychotherapy Scheme (SPS) which provides early opportunities for using psychodynamic psychotherapy and student practice of such to teach doctor-patient relationships and reflection [30].

Further, responsive to recent COVID-imposed exigencies, online forums for RP delivery have been found acceptable by students [24] as have combined group and written reflections [28].

## 4. Discussion

As far as we are aware, this is the first systematic review to capture the evidence base regarding the use of group-based RP to enhance empathy, professionalism, and personal wellbeing amongst medical students. We note that, additionally, other important themes emerged in relation to putative ingredients for a “successful” group as well as innovative delivery models for RP.

Albeit tentative on the basis of a mixed-methods review of studies of variable quality, our findings suggest a range of potentially important learning outcomes of RP in relation to more humanitarian aspects of medicine. For example, when delivered “successfully” (and we as yet do not know what this means), RP may further student understanding of the biopsychosocial context of patients, in particular, conceptualising patients as individuals. Although first proposed by Engel (1977) [40], the biopsychosocial model still remains relevant to medical teaching today as a means of promoting student understanding of the patient’s subjective experience and context, and the effect of psychosocial variables on illness [41].

Promoting understanding of the biopsychosocial context of patients in turn enhances empathy. It is therefore not surprising that there was striking evidence in relation to the effects of RP on empathy borne out in the quantitative studies reviewed [24,25,26,34,35,39]. This is particularly important for more complex patients (often referred to as “heartsink patients” [42]) and also for older patients whose context is often not well-understood by students [28,43]. This has implications for understanding the doctor-patient relationship, and in particular, the resonance or distress associated with specific clinical contexts [27,34]. Such understanding can be akin to psychotherapy concepts of countertransference rendering it not surprising that RP delivery within a psychotherapy teaching programme may bring important insights [30,44]. Understanding the emotive responses to patients is dependent on a process of self-reflection, and awareness of the potential in patients to generate unexpected reactions within the treating clinician. These insights are aligned with the aforementioned benefits of RP [2] and go beyond the more technical skill learning of history taking, use of open questions and other communication skills.

Many of the learning outcomes described here underpin professionalism, which, although is a very broad multidimensional construct, comprises both humanitarianism and the capacity to think critically and reflectively about the doctor-patient relationship in primary service of patient welfare [5,45]. If RP is indeed “instrumental in developing professionalism” then there is an imperative to optimise its teaching [46]. This is all the more so given that the teaching of professionalism is often neglected in medical curricula or relegated to the aforementioned hidden curriculum [47].

We noted that despite the deliberative search based on a commonly used proxy for wellbeing, we found no studies directly examining links either between RP and burnout or in fact any other direct measure of wellbeing. Notwithstanding this, we did observe that RP facilitated connectedness, support and sharing of experiences while mitigating against isolation. Much of these findings appeared to be contingent on the way the RP groups were delivered, bringing us to the salience of some of the additional themes that emerged.

Notwithstanding the lack of operationalisation of what constitutes a “successful” RP group, as educators delivering such groups ourselves, we found some of the additional themes that emerged from the data illuminative from a practical perspective. First, the timing of delivering RP within an undergraduate medical curricula programme is important, and echoed by others seeking to teach professionalism who recognise the need for it to be contextualised during clinical placements to ensure experiential learning and to avoid these important concepts being lost amongst the basic sciences and rote-learning of factual knowledge [48].

Secondly, while there seems to be little doubt as to the importance of the collective experience of RP groups from a learning perspective as stated earlier [2], we would add from the data here that the collective learning of shared emotional experience must be undertaken in a “safe space”. Hearteningly, given the explosion of virtual learning since the COVID pandemic, such a safe space might indeed be achieved in a virtual environment [24].

Thirdly, another question in relation to success arises as to the issue of voluntary versus compulsory groups, particularly relevant to medical educators setting up curricula. Lack et al. (2019), in evaluating whether a compulsory reflective group activity enabled constructive sharing of emotions, noted that students–amongst whom 82% wished to repeat the group RP experience again in their training–reported positive learning experiences, echoed by the facilitators [49]. The non-judgmental format and facilitation with guidance, relevancy and feedback shed light on the potential of structured, collective, and perhaps guided reflection in the educational context. Notwithstanding the finding of this single study, we consider that the issue of compulsory versus non-compulsory groups remains unresolved. Do we focus on the converted, facilitating reflection amongst engaged medical students, who seek to gain the most but perhaps are the least in need of RP? Or do we try to achieve some reflection amongst all students, some of whom will be disengaged or withdraw from groups?

Finally, our data from the identified studies have prompted the question for further examination: is it the group experience, the reflection per se, or the subject focus of the reflection that mediates the “success” of an RP group? This question also needs further testing.

### 4.1. Limitations

The quality of the studies involving quantitative methodology varied widely; with only three of the nine rated as high-quality (score > 80% [16]). The variable quality of these largely data-linked cohort studies does raise the risk of bias and may limit the general applicability of the findings [50]. Notwithstanding this, some of the higher-quality studies emerged in relation to the effect of RP on empathy, potentially lending itself to a quantitative examination of this relationship with a meta-analysis, an opportunity missed by us but perhaps worthy of future studies.

Our strict inclusion and exclusion criteria might have limited our findings. For example, we highlighted the value of group settings for RP based on specific exclusion of solo RP practice, which might have its appeal for students constrained by the fear of exposure and lack of safe space. Moreover, in excluding RP studies involving communications skills (perceiving these as “technical skills”), we may have missed important studies pertaining to RP and empathy mediated by its effect on communication.

Finally, our review is susceptible to language and cultural bias, having excluded non-English papers.

### 4.2. Implications for Future Research

The notable absence of studies examining the relationship between RP and wellbeing, other than non-specific effects addressing loneliness and isolation, highlights an important area for future elucidation. Further, the lack of consistent or comprehensive assessment tools appears to be a consequence of the absence of conceptual or operational clarity regarding what really constitutes “success” or outcomes for these groups beyond empathy change. As far as we are aware, there are no studies examining outcomes related to wellbeing or operationalisation of professionalism.

## 5. Conclusions

Although tentative only, our findings illustrate the long-mooted value of group RP in medical student education. Perhaps the strongest evidence lies in its effect on promoting professionalism and empathy in medical students, being an important target for contemporary medical educators. Further, we have identified some clues as to when and how RP could be delivered, yet to be empirically tested. We join the call to bring to the fore the “hidden curriculum” in medicine and to continually refine and improve its delivery in medical education.

## Figures and Tables

**Figure 1 healthcare-11-01798-f001:**
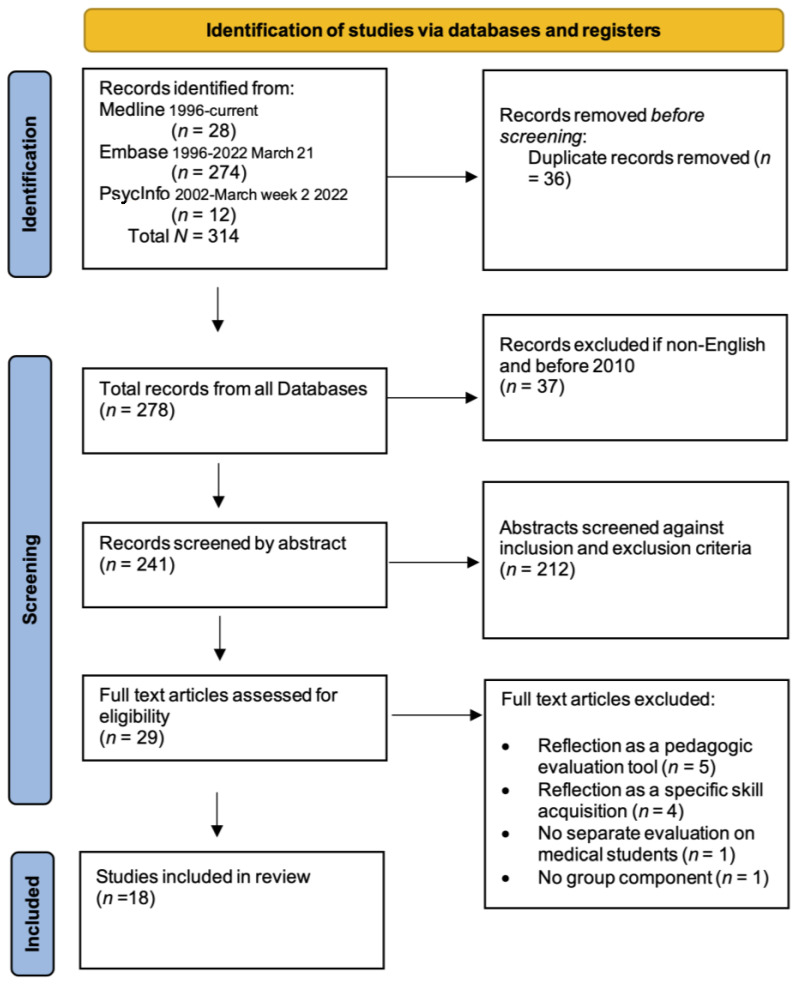
PRISMA flowchart of the results of the systematic review.

## Data Availability

The data collected for this review is available upon request from the corresponding author.

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
