# Peer review of "A Mixed-Methods Systematic Review of Group Reflective Practice in Medical Students"

_healthcare, 2023, doi:10.3390/healthcare11121798_

Round 1

Reviewer 1 Report

Mixed methods reviews offer an innovative framework for generating unique insights related to the complexities associated with health care quality and safety. This is an interesting mixed methods review paper that examine the role of group reflective practice in promoting well-being, empathy and professionalism in medical students.

However, I have the following concerns that has to be addressed for better clarity:

Title: Title should include the term “mixed-methods systematic review” to facilitate identification by potential readers. The authors should use the term “Does” instead of “Can” as “Can” is usually used for efficacy studies.

Abstract: The abstract should identify the paper as a systematic review. It should also specify the inclusion and exclusion criteria for the review, the methods used to assess risk of bias in the included studies, and methods used to present and synthesize results.

The authors have mentioned that they have excluded studies on specific skills. This is quite confusing as Empathy & Professionalism are also regarded as skills.

The abstract at the end should provide the register name and registration number.

Introduction: Overall, the introduction falls short to provide the necessary rationale for conducting the review. It should describe the rationale for this review in the context of existing knowledge and articulate why it is important to do this review (elaborately). Provide the rationale for conducting mixed methods systematic review.

Objectives: the authors statement that “This systematic review aims to explore the literature regarding the use of group-based RP” implies that they are going to include only those studies where group based RP was used specifically to enhance empathy etc., Is that true?

In the research question, the authors need to clarify the “C” comparator component.

Methods: In the search terms I could not find the key terms relevant for well-being. Burnout and well-being are not the same. Not having burnout does not mean well-being.

The authors have mentioned that they have excluded studies on cultural competency. This is quite contradicting with their posed research question when empathy is a component of cultural competency.

I could not find any information on the language restriction as mentioned in the abstract.

The authors need to mention about the validation of their search strategy.

The authors need to mention the team composition with expertise as this review involves both qual and quan synthesis.

Usually methodological qualities are appraised in systematic review. The authors need to clarify the rational for appraising level of evidence.

The authors need to describe the multi-stage process involved in study selection. For example, screening of title, abstract followed by…….

The authors need to clarify whether they adopted convergent integrated approach or convergent segregated approach for their mixed method systematic review synthesis and why so?.

If integrated, the authors should specify the data transformation process that was used to convert the extracted quantitative data into qualitized data to facilitate integration with data extracted from qualitative studies.

If segregated, the approach to the integration of the quantitative and qualitative evidence should be described in as much detail as is reasonably possible.

The authors need to clarify whether their thematic analysis is deductive or inductive.

Results: The authors need to provide information of the studies with regard to the outcomes assessed.

The authors need to provide whether group reflective practice included were formal or informal or delivered as part of routine curriculum.

In 3.4.2. The authors need to provide information on the impact/effect of the RP on empathy.

Being a systematic review, the authors need bring the relevance and the context of the additional themes mentioned in 3.4.4 and 3.4.5 to the posed research questions. Without the connectedness, it would deviate from the systematic review.

The description for other major themes should be as thick as possible.

Discussion: The lines” Albeit tentative only due to the nature of this review, a range of important learning outcomes from implementing RP have emerged here. When delivered ‘successfully’ (and we as yet do not know what this means)” does not seem to fit appropriately particularly for this mixed method systematic review.

Overall, the discussion does not appear to have coherence with the posed research question and the context, as well as the result findings and appears more of a scoping type. I believe barriers and facilitators for success delivery of RP is not the major phenomenon that was explored. I could not find any relevant and focused discussion related to empathy and well-being component.

The discussion should provide information on the issues arising from the conduct of the review, as well as a discussion of the findings of the review and of the significance of the review findings in relation to practice and research.

Author Response

Response to Reviewer 1 Comments

Point 1: Mixed methods reviews offer an innovative framework for generating unique insights related to the complexities associated with health care quality and safety. This is an interesting mixed methods review paper that examine the role of group reflective practice in promoting well-being, empathy and professionalism in medical students.

Response 1: Thank you for these comments.  

However, I have the following concerns that has to be addressed for better clarity:

Point 2: Title: Title should include the term “mixed-methods systematic review” to facilitate identification by potential readers. The authors should use the term “Does” instead of “Can” as “Can” is usually used for efficacy studies.

Response 2: Thank you for making these excellent points. We have now changed the title as suggested. To the issue of implying efficacy, we agree about the inference of the word “Can” (and even “Does” for that matter) and actually preferred the Reviewers’ description of our paper (See Point 1). To this end, we have now removed the question from the title, and changed the PICO question to What is the role of group reflective practice in promoting empathy, professionalism and wellbeing in medical students?

Point 3: Abstract: The abstract should identify the paper as a systematic review. It should also specify the inclusion and exclusion criteria for the review, the methods used to assess risk of bias in the included studies, and methods used to present and synthesize results.

Response 3: Thank you. We have now, as suggested, identified the paper as a systematic review, specified and elaborated on the inclusion and exclusion criteria for the review, and the methods used to assess quality.

Point 4: The authors have mentioned that they have excluded studies on specific skills. This is quite confusing as Empathy & Professionalism are also regarded as skills.

Response 4: We apologise for the confusion caused, this being an excellent point. We have now described the excluded articles as those “focused on technical skill acquisition”, and we have also now conceded this issue as a Limitation: Moreover, in excluding RP studies involving communications skills (perceiving these as “technical skills”), we may have missed important studies pertaining to RP and empathy mediated by its effect on communication.

Point 5: The abstract at the end should provide the register name and registration number.

Response 5: This has now been added.

Point 6: Introduction: Overall, the introduction falls short to provide the necessary rationale for conducting the review. It should describe the rationale for this review in the context of existing knowledge and articulate why it is important to do this review (elaborately). Provide the rationale for conducting mixed methods systematic review.

Response 6: Thank you. We have now elaborated extensively on the argument for the review in the light of existing literature in the Introduction to further justify the rationale for review.

Point 7: Objectives: the authors statement that “This systematic review aims to explore the literature regarding the use of group-based RP” implies that they are going to include only those studies where group based RP was used specifically to enhance empathy etc., Is that true?

Response 7: Yes, that is true. We have also justified this focused approach more clearly now.

Point 8: In the research question, the authors need to clarify the “C” comparator component.

Response 8: We did not use a comparison component, but rather used a modified PICo question, (Population, Interest, Context) which we have now explained and backed up as previous with an appropriate reference (Please see CQUniversity Australia. Framing your research question. CQUniversity Library; 2023. Accessed: 26 May 2023. Available from: https://libguides.library.cqu.edu.au/c.php?g=949210&p=6881572).

Point 9: Methods: In the search terms I could not find the key terms relevant for well-being. Burnout and well-being are not the same. Not having burnout does not mean well-being.

Response 9: Thank you for pointing this out. The justification for using the term “burnout” as a proxy for wellbeing is now provided.  

Point 10: The authors have mentioned that they have excluded studies on cultural competency. This is quite contradicting with their posed research question when empathy is a component of cultural competency.

Response 10: We apologise for this contradiction. We have removed the cultural competency exclusion to avoid any contradiction or confusion. Exactly to the reviewers’ point, we note that enhanced cultural competency emerged as a theme in the Chu et al, 2018 paper.     

Point 11: I could not find any information on the language restriction as mentioned in the abstract.

Response 11: This has now been clarified.

Point 12: The authors need to mention about the validation of their search strategy.

Response 12: We have described our search strategy consistent with PRISMA reporting, using both independent and consensual search and screening strategies, and validating of the search strategy.

Point 13: The authors need to mention the team composition with expertise as this review involves both qual and quan synthesis.

Response 13:  We apologise as we do not understand this comment and are unaccustomed to describing our expertise in a systematic review and do not feel comfortable in doing so. The senior author has published to date over 177 publications including multiple qualitative studies, and multiple systematic reviews, as well as supervising two doctorate systematic reviews, and 1 scoping review.  (see recent: Wand AP, Browne R, Jessop T, Peisah C. A systematic review of evidence-based aftercare for older adults following self-harm. Aust N Z J Psychiatry. 2022 Nov;56(11):1398-1420; Roberts L, Leung KCY, Peisah C. The role of palliative care nurse practitioner in promoting end-of-life care in residential care facilities. J Nursing Education and Practice. 2022; 12 (10) DOI: https://doi.org/10.5430/jnep.v12n10p7; also Xie Y, Hamilton M, Peisah C, Anstey KJ, Sinclair C. Navigating Community-Based Aged Care Services From the Consumer Perspective: A Scoping Review. Gerontologist. 2023 Apr 29:gnad017. doi: 10.1093/geront/gnad017. Epub ahead of print. PMID: 37120292.

Point 14: Usually methodological qualities are appraised in systematic review. The authors need to clarify the rational for appraising level of evidence.

Response 14: The rationale for appraising methodological qualities is extensively described in Section 2.5 Quality Assessment” It is now additionally described in the abstract as requested by the Reviewer.

Point 15: The authors need to describe the multi-stage process involved in study selection. For example, screening of title, abstract followed by…….

Response 15: Please refer to Figure 1 PRISMA flowchart for detailed multi-stage process with exclusion rationales. We prefer not to double up description of methodology in both Text and Tables. 

Point 16: The authors need to clarify whether they adopted convergent integrated approach or convergent segregated approach for their mixed method systematic review synthesis and why so?.

Response 16: Thank you for your most astute comments here. We used a convergent integrated approach for our mixed-methods systematic review, combining both forms of data (i.e. qualitative and quantitative) into a single mixed methods synthesis, codifying both forms of data using thematic analysis. We have now elaborated on and clarified this in the Methods.

Point 17: If integrated, the authors should specify the data transformation process that was used to convert the extracted quantitative data into qualitized data to facilitate integration with data extracted from qualitative studies.

Response 17: See response Point 16.

Point 18: If segregated, the approach to the integration of the quantitative and qualitative evidence should be described in as much detail as is reasonably possible.

Response 18: Not applicable – see above.

Point 19: The authors need to clarify whether their thematic analysis is deductive or inductive.

Response 19: This has now been clarified.

Point 20: Results: The authors need to provide information of the studies with regard to the outcomes assessed.

Response 20: Please refer to Appendix, Table A1, for detailed information and outcomes of the studies reviewed.

Point 21: The authors need to provide whether group reflective practice included were formal or informal or delivered as part of routine curriculum.

Response 21: This has now been added and clarified.

Point 22: In 3.4.2. The authors need to provide information on the impact/effect of the RP on empathy.

Response 22: This has now been clarified and provided both in the Results and reflected upon in the Discussion.

Point 23: Being a systematic review, the authors need bring the relevance and the context of the additional themes mentioned in 3.4.4 and 3.4.5 to the posed research questions. Without the connectedness, it would deviate from the systematic review.

Response 23: Thank you. The emergence, context and relevance of the additional themes has now been explicitly stated in the Abstract, the Results, and the Discussion. 

Point 24: The description for other major themes should be as thick as possible.

Response 24: We have now expanded on the major themes.

Point 25: Discussion: The lines” Albeit tentative only due to the nature of this review, a range of important learning outcomes from implementing RP have emerged here. When delivered ‘successfully’ (and we as yet do not know what this means)” does not seem to fit appropriately particularly for this mixed method systematic review.

Response 25: We have adopted a tentative stance here as we want to engage readers to explore with us the ingredients for implementing ‘successful’ RP groups. As the paragraphs continue, the themes and content of this review and lessons learned referring to other existing studies emerge. We then refer back to the PICo question at the end, and prompt further research to answer the questions posed by the review.

Point 26: Overall, the discussion does not appear to have coherence with the posed research question and the context, as well as the result findings and appears more of a scoping type. I believe barriers and facilitators for success delivery of RP is not the major phenomenon that was explored. I could not find any relevant and focused discussion related to empathy and well-being component.

Response 26: We have made significant modifications to encompass a more coherent Discussion.

Point 27: The discussion should provide information on the issues arising from the conduct of the review, as well as a discussion of the findings of the review and of the significance of the review findings in relation to practice and research.

Response 27: Please refer to 4.1 Limitations for information on the issues arising from the conduct of the review. Please refer to 4.2 Implications for future research and 5.0 Conclusion for comments on the significance of the review.  

Reviewer 2 Report

Very interesting topic, the researchers did a great job to review the literature regarding the use of group reflective practice in fostering empathy, wellbeing, and professionalism in medical students. However, the article could be strengthened through: 

Introduce every acronym before using it in the text. The first time you use the term, put the acronym in parentheses after the full term, like COVID 19 in line 51.  

Add more towards scope of the problem in introduction section. 

Remove authors names, like CP, KCYL in line 98. 

Add objectives of the paper at the end of introduction. 

Add more current references from the literature.  

Add more towards the result of the review with regard to the outcomes assessed. 

Expand discussion section.  

Author Response

Response to Reviewer 2 Comments

Point 1: Very interesting topic, the researchers did a great job to review the literature regarding the use of group reflective practice in fostering empathy, wellbeing, and professionalism in medical students.

Response 1: Thank you very much.

However, the article could be strengthened through: 

Point 2: Introduce every acronym before using it in the text. The first time you use the term, put the acronym in parentheses after the full term, like COVID 19 in line 51.  

Response 2: Thank you for your suggestion. We have decided to retain the term COVID as it is widely accepted in the scientific literature and used by international authorities such as the World Health Organisation. This is also to enhance readability, so that the flow of the sentence is not disrupted for the reader. We have checked all other acronyms and all their full terms are mentioned when first introduced.

Point 3: Add more towards scope of the problem in introduction section. 

Response 3: Thank you. We have now elaborated extensively on the scope of the problem and the argument for the review in the light of existing literature in the Introduction, to further justify the rationale for review.

Point 4: Remove authors names, like CP, KCYL in line 98. 

Response 4: We have done this as requested in all references to the two authors.  

Point 5: Add objectives of the paper at the end of introduction. 

Response 5: We end the Introduction with the PICo question of this systematic review, which also serves as the objectives of the paper, in a Systematic Review, as per the PRISMA 2020 checklist. Please also see Page MJ, Moher D, Bossuyt PM, Boutron I, Hoffmann TC, Mulrow CD, et al. PRISMA 2020 explanation and elaboration: updated guidance and exemplars for reporting systematic reviews. BMJ 2021;372:n160. doi: 10.1136/bmj.n160

Point 6: Add more current references from the literature.  

Response 6: We considered that 50 references, of which 18 were examined in detail serving as the basis for the review, chosen from a pool of 314 papers, served as sufficient basis of this review, and was commensurate with other reviews, including our own.   

Point 7: Add more towards the result of the review with regard to the outcomes assessed. 

Response 7: Please refer to Appendix, Table A1 with regard to details of the outcomes assessed. The table is also referenced under Results.

Point 8: Expand discussion section.  

Response 8: This has been expanded as requested.

Reviewer 3 Report

This manuscript presents findings from a review of literature regarding reflective practice among medical students.  Thematic analysis was used with eighteen articles, which had been included from a PRISMA approach to systematic review.  

Themes were identified in the review, including empathy for patients, well-being of medical students, ingredients for successful group reflective practices, and innovative delivery methods for reflective practice.  Discussion included the flow of empathy from the biopsychosocial model, professionalism in the doctor-patient relationship, and conditions for successful delivery of group reflective practice among medical students.  Limitations were identified and discussed.

This review identifies important concepts for improving the doctor-patient relationship and also identifies self-care practices for medical students.  By bringing these concepts out of the "hidden curriculum" of medical training and into the "open curriculum," there is potential to improve medical training and therefore patient care.  There is great value in this material for individual physicians, patients, and society.  The manuscript was well-written and easy to follow. 

Author Response

Response to Reviewer 3 Comments

This manuscript presents findings from a review of literature regarding reflective practice among medical students.  Thematic analysis was used with eighteen articles, which had been included from a PRISMA approach to systematic review.  

Themes were identified in the review, including empathy for patients, well-being of medical students, ingredients for successful group reflective practices, and innovative delivery methods for reflective practice.  Discussion included the flow of empathy from the biopsychosocial model, professionalism in the doctor-patient relationship, and conditions for successful delivery of group reflective practice among medical students.  Limitations were identified and discussed.

This review identifies important concepts for improving the doctor-patient relationship and also identifies self-care practices for medical students.  By bringing these concepts out of the "hidden curriculum" of medical training and into the "open curriculum," there is potential to improve medical training and therefore patient care.  There is great value in this material for individual physicians, patients, and society.  The manuscript was well-written and easy to follow. 

Response: Thank you very much for your positive feedback. We have also integrated some of your valuable insights.

Reviewer 4 Report

The paper is well-written and concise. To add to gauging interest among researchers, I suggest:

Line 15: this explanation is not relevant to be mentioned in the  Abstract.

Lines 16-18: “Settings included United States(6), United Kingdom(3),Australia(3), France(2), Taiwan(2), Germany(1), and Ireland(1).”  This information is not relevant here and could be substituted for other more elucidative

Lines 21-22: These statements could be better replaced by what is suggested straight forward here.

Lines 55-56: This question, if put at the beginning of the Abstract, would gauge the interest of other researchers.

Lines 297-298: This exposition is very elucidating about the contribution of this systematic review and should be emphasized in the Abstract

Lines 307-310: this information is very appealing and should be included in the Abstract.

Author Response

Response to Reviewer 4 Comments

Point 1: The paper is well-written and concise.

Response 1: Thank you very much.

To add to gauging interest among researchers, I suggest:

Point 2: Line 15: this explanation is not relevant to be mentioned in the Abstract.

Response 2: We appreciate this suggestion. After considering different reviewers’ comments and consulting other published systematic reviews, we have retained this to succinctly capture the key exclusion criteria.

Point 3: Lines 16-18: “Settings included United States(6), United Kingdom(3),Australia(3), France(2), Taiwan(2), Germany(1), and Ireland(1).”  This information is not relevant here and could be substituted for other more elucidative

Response 3: From our previous experience undertaking reviews, as well as observing other reviews, this information regarding the diversity of settings is traditionally provided.   

Point 4: Lines 21-22: These statements could be better replaced by what is suggested straight forward here.

Response 4: These have now been replaced as suggested.

Point 5: Lines 55-56: This question, if put at the beginning of the Abstract, would gauge the interest of other researchers.

Response 5: Thank you for this suggestion. We have put the PICo question: “the role of group RP in fostering empathy, wellbeing, and professionalism in medical students” at the beginning of the abstract.   

Point 6: Lines 297-298: This exposition is very elucidating about the contribution of this systematic review and should be emphasized in the Abstract

Response 6: Thank you. This has now been added to the Abstract.

Point 7: Lines 307-310: this information is very appealing and should be included in the Abstract.

Response 7: Thank you. This has now been included in the Abstract.

Round 2

Reviewer 1 Report

The authors have addressed most of the concerns diligently.

I have few minor suggestions:

For point 13:  A review team may comprise experienced systematic reviewers, information specialists, statisticians, and content experts. In the current study, the authors can provide details about the composition of their team and the specific expertise each member brings, focusing on their ability to handle qualitative and quantitative studies as well as their integration in the review process. This will add value to their review.

The description of major themes can still be expanded to provide a detailed account.

Author Response

Response to Reviewer 1 Comments

The authors have addressed most of the concerns diligently.

Authors’ response: Thank you for this comment.

I have few minor suggestions:

For point 13:  A review team may comprise experienced systematic reviewers, information specialists, statisticians, and content experts. In the current study, the authors can provide details about the composition of their team and the specific expertise each member brings, focusing on their ability to handle qualitative and quantitative studies as well as their integration in the review process. This will add value to their review.

Authors’ response: Thank you for this suggestion and advice. We have now provided a separate section in the Method titled “2.4 Review team”:

“The review team comprised experienced systematic reviewers, with qualitative and quantitative research experience, as well as being content experts responsible for delivering RP groups.  The lead author, an advanced psychiatry trainee with expertise in delivering RP groups and undertaking systematic reviews, undertook the database searches, screening and integration. The second author, who is a senior psychiatrist with extensive experience undertaking qualitative research and systematic reviews, as well as delivering RP groups, assisted with screening and the thematic analysis.”

The description of major themes can still be expanded to provide a detailed account.

Authors’ response: Thank you for this suggestion. We agree that this provides more fleshing out of the data. We have now expanded on the description of the major themes to provide more detail (see 3.4 Synthesis of Results).